# Understanding the Effectiveness of Cross-Domain Contrastive Unsupervised Domain Adaptation

**Viacheslav Sinii[1], Adín Ramírez Rivera[2], Adil Khan[1,3]**
[1]Innopolis University, [2]University of Oslo, [3]University of Hull

## Abstract

Unsupervised domain adaptation helps to transfer learned tasks from a source to a target domain in the lack of labeled data. Recently, contrastive learning has shown promising results in this setup. However, there are limitations on the performance due to unbalanced objectives between the self-representation and the adaptation tasks. We show that pre-training choices and hard negative mining provide on average 20% improvement of accuracy and successfully pair contrastive learning and unsupervised domain adaptation.

## 1 Introduction

Unsupervised domain adaptation (UDA) aims to improve the performance of a machine learning model on a target domain whose characteristics are different from those on the source domain in which it is trained. This is done without using any labeled data from the target domain by learning a transferable feature representation that captures the shared information between the two domains, which is especially helpful in low data regimes. Existing approaches give theoretical bounds on target error (Ben-David et al., 2006; 2010); suggest aligning probability distributions using carefully designed metrics (Gretton et al., 2006; Long et al., 2015; 2017; Zellinger et al., 2017); and, recently, contrastive learning was used for domain adaptation (Chen et al., 2020; He et al., 2020; Kang et al., 2019; Khosla et al., 2020; Toldo et al., 2021; Wang et al., 2022) (Appendix F). However, improper use of these techniques can lead to suboptimal results. Our work examines the limitations of the contrastive approach for UDA introduced by Wang et al. (2022). We show that their method underperforms the naive model that has no domain adaptation. We found that this degradation is due to the unbalanced effect of contrastive and classification objectives on model training where domain adaptation shadows the main task of classification. Through extensive experiments, we found that pre-training choices and explicit hard negative sampling can improve model performance.

## 2 Background

The formal definition of UDA setting can be found in Appendix A.1.

Wang et al. (2022) employ Cross-Domain Contrastive Loss (*CDCL*) for learning domain-invariant class-aware features. We describe their methodology in detail in Appendix A. This approach has caught our attention since it raised several important questions:

- How will additional objective, i.e. domain adaptation expressed in contrastive loss, affect the main task (classification)?
- The pseudo-labeling will inevitably produce incorrect labels, especially at the beginning of training. How will this affect model performance?

Thus, the focus of our study is understanding the impact of contrastive loss on model training.

---

Source code: `https://github.com/ummagumm-a/cross_domain_contrastive_uda`.

Table 1: Target accuracy (%) on Office-31 (ResNet-50) and VisDA2017 (ResNet-101) for unsupervised domain adaptation.

| Method | Office-31 | | | | | | VisDA | Avg. |
|---|---|---|---|---|---|---|---|---|
| | A→ W | A→ D | W→ A | W→ D | D→ A | D→ W | | |
| No adaptation | 69.9 ± 8.2 | 79.0 ± 9.7 | 63.0 ± 9.1 | 95.1 ± 2.8 | 62.1 ± 3.5 | 88.9 ± 6.9 | 51.5 ± 3.4 | 72.8 |
| CDCL | 57.0 ± 18.2 | 58.5 ± 3.3 | 54.4 ± 14.7 | 55.3 ± 25.9 | 54.1 ± 9.0 | 68.3 ± 16.6 | 60.3 ± 10.6 | 58.3 |
| Random Sampling | 57.3 ± 5.3 | 60.7 ± 11.3 | 53.1 ± 3.1 | 59.6 ± 4.0 | 48.3 ± 9.0 | 73.1 ± 19.5 | 62.5 ± 25.2 | 59.2 |
| Pretraining | 79.1 ± 6.0 | 75.9 ± 8.8 | 64.2 ± 1.0 | 90.6 ± 3.2 | 67.5 ± 2.2 | 93.4 ± 0.5 | 61.4 ± 5.5 | 76.0 |
| Explicit Negative Sampling | 78.7 ± 4.9 | 84.9 ± 8.6 | 68.6 ± 5.9 | 94.9 ± 3.9 | 69.2 ± 4.7 | 92.1 ± 3.3 | 70.7 ± 21.2 | 79.9 |
| Implicit Negative Sampling | 74.8 ± 11.6 | 78.7 ± 5.9 | 68.9 ± 3.6 | 82.4 ± 13.7 | 70.7 ± 5.4 | 89.2 ± 8.9 | 71.2 ± 2.0 | 76.6 |

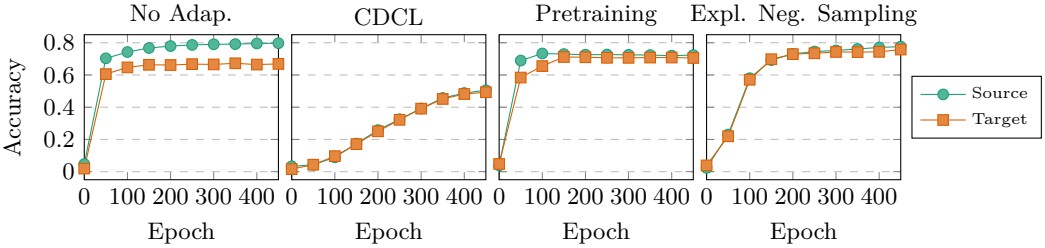

Figure 1: The accuracy of the baseline model with no domain adaptation. The gap between source and target accuracies signifies a misalignment between the two domains.

## 3   FINDINGS

Compared to the *Baseline* model with no domain adaptation (Appendix B), *CDCL* resulted in two well-aligned domains (Figure 1); the gap between the source and the target accuracies is negligible. However, accuracy for both domains dropped significantly, and the convergence rate slowed down. Overall, *CDCL* achieved a reduced domain gap by sacrificing accuracy (Table 1).

We believe that this result is due to the overwhelming impact of the contrastive loss, which can be seen by examining gradient norms w.r.t. to each of the two losses. Apart from that, the confidence of the predictions of the baseline model stays at low values of approx. 3% (Appendix C).

Since classification is overshadowed during training, we let the model to *pre-train* on the source domain before activating the contrastive loss (Appendix B). Results in Table 1 and Figure 1 show that this technique improved the performance by 18% on average.

Additionally, we discovered that explicit hard *negative sampling* helps to balance the two losses, thus shifting focus on classification while still being able to align two domains. Table 1 indicates an average of 20% improvement over CDCL. To validate the positive impact of explicit hard negative sampling, we quantitatively show that neither random sampling nor implicit negative sampling lead to the same improvement (Table 1; Appendix D.

Finally, we found that CDCL is resistant to noise and is not affected by incorrectly labeled target samples (Appendix E). The details of our experimental setup can be found in Appendix B.

## 4   CONCLUSION

Our experiments demonstrated that contrastive loss overshadows the classification loss and therefore techniques for their balancing are required. Pretraining of the model ensures the reasonable performance of the classifier before the start of domain adaptation. However, explicit hard-negative sampling may be a preferable method since it improves the classification accuracy and domain alignment in an easier manner, outperforming other approaches.

ACKNOWLEDGEMENT

This research has been financially supported by The Analytical Center for the Government of the Russian Federation (Agreement No. 70-2021-00143 dd. 01.11.2021, IGK 000000D730321P5Q0002)

URM STATEMENT

The authors acknowledge that at least one key author of this work meets the URM criteria of ICLR 2023 Tiny Papers Track.

CONTACT INFORMATION

Viacheslav Sinii - `v.sinii@innopolis.university`
Adín Ramírez Rivera - `adinr@uio.no`
Adil Khan - `a.khan@innopolis.ru, a.m.khan@hull.ac.uk`

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

## A   CDCL

This section describes Cross-Domain Contrastive Loss (CDCL) proposed by Wang et al. (2022).

### A.1   Setup

In UDA, the aim is to adopt a model trained on a labeled source domain to perform well on other unlabeled domains. Formally, the source dataset is defined as a set of $N_s$ tuples $(X_s, Y_s) = \{(x_s^i, y_s^i) : i \in [1; N_s]\}$ and target dataset is defined as a set of $N_t$ samples $X_t = \{x_t^i : i \in [1; N_t]\}$. The samples in $X_s$ and $X_t$ are drawn from the source and target distributions $D_s$ and $D_t$. Also, $x_s$ and $x_t$ belong to the same fixed set of $M$ categories, i.e., $y_s^i \in \{0, \ldots, (M-1)\}$. Then, the task is to construct a model $f_t : X_t \rightarrow Y_t$ trained on $X_s \cup X_t$ which consists of feature encoder $g : X_t \rightarrow \mathbb{R}^d$ and classification head $h : \mathbb{R}^d \rightarrow Y_t$, where d is the dimensionality of feature space. $z_s^i$ and $z_t^j$ are defined to be L2-normalized feature encodings of $x_s^i$ and $x_t^j$.

## A.2 PSEUDO-LABELING

The alignment of feature distributions on the class level is crucial for the high performance of a model. However, class information for the target domain is unavailable and requires approximation. That can be achieved with a technique called pseudo-labeling which uses K-means clustering to infer class information. This approach is based on the assumption that feature encodings of objects from the same class are likely to locate close to each other irrespective of their domain. Cluster centers are initialized to class centroids from the source domain, and the algorithm runs until convergence. Then, samples assigned to a cluster are pseudo-labeled as belonging to the class that was used to initialize this cluster. To reduce the labeling noise, all samples located far from cluster centers are removed; specifically, the samples are removed when the cosine similarity between their features and cluster centers is less than a threshold $d$.

## A.3 CONTRASTIVE LOSS

Contrastive loss is generally utilized to produce feature space in which 'similar' samples (positive pairs) are located closer to each other, and 'dissimilar' samples (negative pairs) are far from each other. In the context of UDA, the contrastive loss is used to ensure that samples from the same class are mapped to the same location in the feature space irrespective of their domain. Therefore, a positive pair is defined as samples that belong to the same class and are from different domains. Similarly, a negative pair consists of samples from different classes and different domains. In each training step, one batch is sampled from each domain, and each target sample in a batch acts as an anchor. Then, the formulation of Cross-Domain Contrastive Loss is:

$$L_{\text{CDC}}^{t,i} = -\frac{1}{|P_s(\hat{y}_t^i)|} \sum_{p \in P_s(\hat{y}_t^i)} \log \frac{\exp(z_t^i z_s^p / \tau)}{\sum_{j \in I_s} \exp(z_t^{iT} z_s^j / \tau)} \tag{A.1}$$

where $i$ is the index of the anchor sample, $I_s$ is the set of source samples in a mini-batch, $P_s(\hat{y}_t^i) = \{k \mid y_s^k = \hat{y}_t^i\}$ indicates the set of positive samples from the source domain that share the same label with the target anchor $x_t^i$, and $\tau$ is a temperature parameter.

Similarly, source samples are used as anchors to compute $L_{\text{CDC}}^{s,i}$. Combining $L_{\text{CDC}}^{t,i}$ and $L_{\text{CDC}}^{s,i}$ gives the final definition of the Cross-Domain Contrastive Loss:

$$L_{\text{CDC}} = \sum_i^{N_s} L_{\text{CDC}}^{s,i} + \sum_j^{N_t} L_{\text{CDC}}^{t,j}. \tag{A.2}$$

## A.4 CLASSIFICATION

Separately from two batches sampled for contrastive loss, another batch is sampled from the source dataset to train the classifier. The outputs of classification head $h$ are used to evaluate classification performance, and Cross-Entropy loss is used as the optimization objective:

$$L_{\text{CE}} = -\sum_i^M y_i log(p_i) \tag{A.3}$$

where $y_i$ is 1 when a sample $x$ is labeled as class $i$ and 0 otherwise; $p_i$ is a probability of a sample $x$ belonging to class $i$ that comes from model predictions.

## A.5 FINAL OPTIMIZATION OBJECTIVE

Classification and contrastive losses are combined with a weighting parameter $\lambda$ to define the final optimization objective:

$$\min_\theta \quad L_{\text{CE}}(\theta; D_s) + \lambda L_{\text{CDC}}(\theta; D_s, D_t). \tag{A.4}$$

## B EXPERIMENTAL SETUP

In our experiments, for fair comparison we took the same model architecture and datasets as Wang et al. (2022).

### B.1 DATASETS

**Office-31** contains 31 classes of objects typically encountered in an office environment. It is split into three domains—'amazon' with images taken with a good camera on a white background; 'dslr' with images taken in an office with a high-resolution camera on various backgrounds; and 'webcam' with images also collected in an office environment but with a low-resolution camera exhibiting some noise.

**VisDA2017** is a large-scale dataset having 12 classes. In our experiments, we evaluated model performance on the synthesis-to-real adaptation task.

We performed analysis on the 'amazon-webcam' pair. Other datasets were used for final evaluation; the results are presented in Table 1.

### B.2 IMPLEMENTATION DETAILS

**CDCL.** We used a separate backbone for each dataset - ResNet50 for Office-31 and ResNet101 for VisDA, both are pretrained on ImageNet (Deng et al., 2009). In addition, we used Domain-Specific BatchNorm (Chang et al., 2019), an L2-normalization component for outputs, and a single fully-connected layer with a bias component disabled as the classification head.

We used SGD with a momentum of 0.9 for model optimization with the learning rate set to $1e^{-3}$ for the backbone and $1e^{-2}$ for the classification head. Learning rate scheduler: $\eta = \eta_0(1 + 10p)^{-b}$, where $p$ linearly increases from 0 to 1 and $b$ was set to 0.75 (for Office-31) and to 2.25 (for VisDA). All images were resized to $(224, 224)$ shape, and pixel values are normalized to mean 0 and variance 1. The number of epochs is 500.

For contrastive loss: temperature parameter $\tau = 1.0$ and weighting parameter $\lambda = 1.4$.

For a reliable estimate of model performance, we averaged the results over four dataset splits produced by StratifiedKFold.

Changes described below were made with respect to the settings of **CDCL**.

**No Adaptation (Baseline).** This model was trained on the source domain with the target domain used only for evaluation. This was achieved by setting weighting parameter $\lambda = 0$.

**Pretraining.** The model was trained for 100 epochs on the source dataset; then the contrastive loss was activated.

**Explicit Negative Sampling.** Expression A.1 was substituted for expression D.1 with threshold parameter $s_\alpha = 0.5$.

**Implicit Negative Sampling.** Temperature parameter $\tau = 0.05$. For Office-31 weighting parameter $\lambda = 0.1$.

**Random Sampling.** In each training step, we chose random negative pairs in the same amount as the Explicit Negative Sampling thresholding would give.

**Obtaining gradients w.r.t. to each loss.** We calculated the gradients w.r.t. to each component in the expression A.4 separately - $L_{CE}(\theta; D_s)$ and $\lambda L_{CDC}(\theta; D_s, D_t)$. Then, we treated gradients for all the weights as a single vector and calculated its L2-norm.

**Contrastive-only model** The feature encoder was updated only with gradients w.r.t. to the $L_{CDC}$. Since classification head $h$ does not participate in calculation of contrastive loss, its weights were not updated by $L_{CDC}$. Therefore, classification loss $L_{CE}$ was calculated, but its impact was limited only to the classification head.

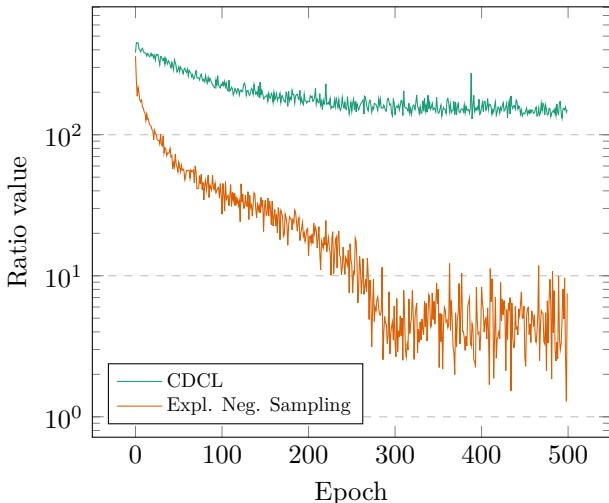

Figure C.1: Ratio between gradient norms w.r.t. contrastive and classification loss. The rapid decrease of the ratio for Explicit Negative Sampling training strategy highlights its ability to balance the two losses.

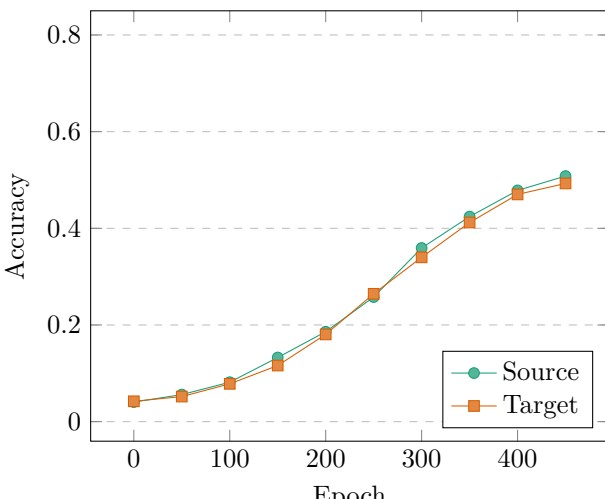

Figure C.2: Performance of the Contrastive-only model. This figure resembles the Figure 1, which means that for default settings the classification loss has no effect on weight update.

## C   ANALYSIS OF CDCL

The fact that the model succeeds in aligning two domains well but sacrifices the accuracy can signify that the two losses are combined disproportionally. Indeed, Figure C.1 shows that for CDCL the gradients w.r.t. to contrastive loss dominate the gradients w.r.t. classification loss, so the model gets too concentrated on domain alignment, and classification loss effectively has no impact on weight update. Figure C.2 shows the model performance with classification loss impacting only task-specific classification head, i.e., gradients w.r.t. classification loss do not affect the weights of feature encoder. The curves produced by *CDCL* (Figure 1) and the contrastive-only model are almost identical, which signifies that contrastive loss indeed greatly overweights the classification loss. The details of these experiments are described in Appendix B.

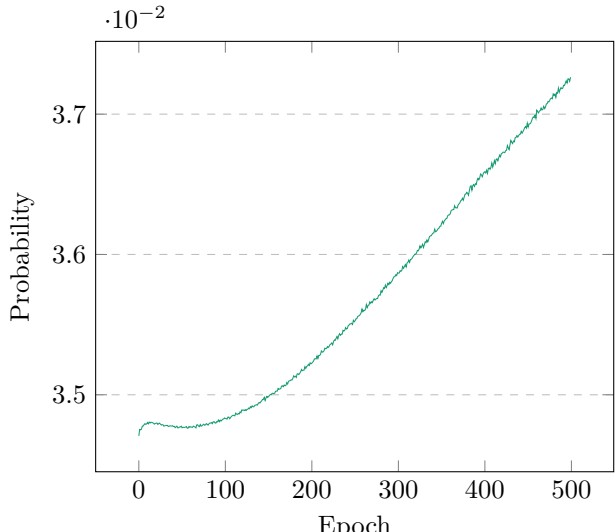

Figure C.3: The mean probability of predicted class in each epoch for CDCL. Though overall the model becomes more confident in its predictions throughout training, the confidence stays at low values.

Also, we found that the model is not confident in its predictions. Figure C.3 shows that though the mean probability of predicted class grows over time, it remains at low values of approx. 3%.

## D  EXPLICIT HARD NEGATIVE SAMPLING

Explicit Hard Negative Sampling is intended to make the model focus only on the hardest examples, thus increasing the efficiency of training. Explicit Hard Negative Sampling is defined as:

$$L_{\text{CDC}}^{t,i} = -\frac{1}{|P_s(\hat{y}_t^i)|} \sum_{p \in P_s(\hat{y}_t^i)} \log \frac{\exp(z_t^i z_s^p / \tau)}{\sum_{j \in I_s, z_t^{iT} z_s^j > s_\alpha} \exp(z_t^{iT} z_s^j / \tau)}, \tag{D.1}$$

where $s_\alpha$ is the threshold parameter for filtering hard negatives.

Since contrastive loss greatly overshadows classification, we decided to decrease its impact on the gradients of the parameters by limiting the set of samples on which it operates. Figure C.1 shows that with this approach the impact of contrastive loss steadily decreases as the set of hard negatives becomes smaller. Importantly, the best performance was achieved when the model was fed with the hardest examples. Our experiments showed that random sampling either has no effect or leads to worse performance (Table 1). Alternatively, Negative Sampling can be performed in an implicit way by lowering the value of temperature parameter $\tau$. Wang & Liu (2021) describe this behavior of contrastive loss and show the superiority of Explicit Negative Sampling. Our experiments also indicate that on average Implicit Negative Sampling gives worse results (Table 1), which is in line with the work of Wang & Liu (2021).

## E  REMOVING INCORRECTLY LABELED SAMPLES

Our experiments showed that on average approx. 15% of training target samples were mislabeled for the '$A \to W$' dataset pair. Despite this, the methodology suggested by Wang et al. (2022) is resistant to incorrect pseudo-labels present in the training dataset. To support this conclusion, we conducted the following experiment: we used the original CDCL setup as described in Appendix B, but after each pseudo-labeling procedure we filtered out samples for which $y_{real} \neq y_{pseudo}$. In this way, the model received only correctly labeled samples throughout training. Figure E.1 shows the performance of this model—the curves almost

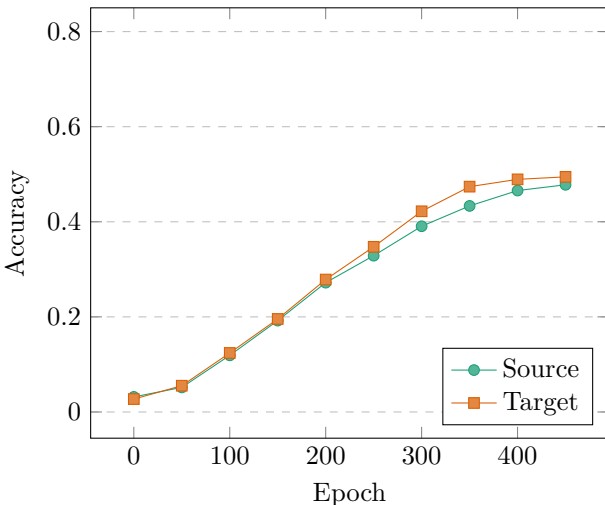

Figure E.1: The performance of CDCL trained with removal of incorrectly pseudo-labeled target samples. This figure resembles Figure 1, which indicates that CDCL is resistant to pseudo-labeling noise.

exactly repeat the ones on Figure 1 (b) showing that the presence of noise in training data does not affect the quality of the model.

## F  RELATED WORK

Previous studies (Ben-David et al., 2006; 2010) have provided theoretical bounds on target error in terms of source error, domain divergence, and shared expected loss. Therefore, many UDA works focus on minimizing domain divergence; they can be broadly categorized into two paradigms.

The first paradigm is based on adversarial approach. For instance, Ben-David et al. (2006) propose using a binary classifier to distinguish between two domains as a way of measuring domain divergence. This idea leads to the adversarial model proposed by Ganin & Lempitsky (2015), which extracts domain-invariant features by reversing gradients from the domain discriminator to the feature extractor. Another example is the use of Generative Adversarial Networks (GANs) to transform source images to look like they are drawn from the target distribution (Bousmalis et al., 2017).

The second paradigm is based on aligning probability distributions of features between the source and target domains. This approach is motivated by the study of Yosinski et al. (2014) who showed that transferability decreases in higher layers of a neural network, and introduced the idea of minimizing discrepancy. Several studies have proposed metrics for aligning probability distributions, e.g. Maximum Moment Discrepancy (MMD) (Gretton et al., 2006; Long et al., 2015; 2017), Central Moment Discrepancy (CMD) (Zellinger et al., 2017), and aligning covariance matrices of features from the top level of the feature extractor (Sun & Saenko, 2016; Sun et al., 2016).

Some studies have attempted to address misalignment on the class level. For example, Saito et al. (2018) trained two classifiers on labeled source data and minimized disagreement between their predictions for target data. Kang et al. (2019) suggest a Contrastive Domain Discrepancy (CDD) metric for distributions conditioned on class information. Saenko et al. (2010) attempted to find a transform into a latent space where the source and target domains look similar while preserving class information. However, these approaches require labeled examples for each class in the target data.

In UDA, labels for target data are not available but can be approximated by pseudo-labeling (Kang et al., 2019; Liang et al., 2020; Toldo et al., 2021). For example, Kang et al. (2019)

used K-means algorithm to cluster target data; initial cluster centers were set to centroids of classes calculated on the source data.

Our work is inspired by Wang et al. (2022), who applied contrastive learning to minimize cross-domain distances between samples of the same category.

