# OpenReview forum: "Understanding the Effectiveness of Cross-Domain Contrastive Unsupervised Domain Adaptation"
_ICLR.cc/2023/TinyPapers — Submitted to Tiny Papers @ ICLR 2023_

### Official Review · Reviewer_YJCb · 2023-03-26

**Confidence:** 4

**Summary Of Contributions:**

This paper considered domain adaptation. Sepcifcally, We show that pre-training choices and hard negative mining provide consistent improvements to successfully pair contrastive learning and unsupervised domain adaptation

**Rating:**

Clear, Correct, and Reproducible (CCR): a submission which meets the reviewing criteria

**Strengths And Weaknesses:**

This paper is generally quite clear (I really like figure 1). The presented technique seems correct and reasonably fine. ThusI think it is a good paper.

I do not have significant concerns. Due to the page limit, some parts still seem unclear. This could be addressed in the final version.





**Suggested Changes:**

I would suggest reporting statistical test and a variance in Table 1.

---

> ### Author Response · Authors · 2023-05-23
> **Answer to Suggested Changes**
>
> Dear Reviewer YJCb,
>
> We thank you for the time it took to review our paper. Your feedback is invaluable for improving the quality of our work. We have considered your suggestions and added variance of the model performance to Table 1.
>
> Thank you for your time and consideration.

---

### Meta-Review · Area_Chair_CAVK · 2023-04-04

**Recommendation:** Invite to present
**Confidence:** 4

**Metareview:**

Clear paper with solid results.

**Summary:**

 The authors demonstrate that employing re-training strategies and hard negative mining consistently enhances performance, effectively combining contrastive learning with unsupervised domain adaptation.

**Reason For Not Giving A Higher Recommendation:**

The paper is well-written. The empirical findings are interesting. However, I do not see any outstanding reason for a highlight.

**Reason For Not Giving A Lower Recommendation:**

Practical problem and interesting results.

---

### Decision · Program_Chairs · 2023-04-10

Invite to present